# OpenReview forum: "Monitoring LLM-based Multi-Agent Systems Against Corruption Attacks via Node Evaluation"
_ICLR.cc/2026/Conference — Submitted to ICLR 2026_

### Official Review · Reviewer_hwcL · 2025-10-27

**Soundness:** 3
**Presentation:** 3
**Contribution:** 3
**Rating:** 8
**Confidence:** 3

**Summary:**

This paper tackles the security challenge of corruption propagation in Large Language Model(LLM)-based Multi-Agent Systems(MAS) by proposing a  dynamic defense paradigm. The approach models the MAS as a directed acyclic graph (DAG). Its core technique involves evaluating the contribution of each agent by first assigning a contribution score (-1,0,1) to each communication edge using an LLM independent of the MAS, and then propagating these scores backward through the graph via a PageRank-like algorithm. Agents whose contribution scores significantly deviate from the group are identified as malicious nodes and subsequently isolated by pruning their outgoing communication edges.
Through experiments, the authors demonstrate that their method outperforms several existing defense baselines in both task accuracy and malicious agent detection rate, showing particularly prominent performance in dynamic attack scenarios.

**Strengths:**

- Originality: The paper introduces a dynamic defense paradigm, using a signed graph and backpropagation to evaluate node contributions, which is a creative approach.
- Quality: The method is well-structured and validated through comprehensive experiments across multiple MAS architectures, benchmarks, and attack types.
- Clarity: The paper is clearly written, with a logical flow that effectively introduces the problem, methodology, and contributions. The presentation of figures and numerical results is organized and easy to follow.

**Weaknesses:**

W1 Subjective and Coarse-Grained LLM Scoring: The core mechanism relies on an external LLM to assign a simplified ternary contribution score (-1,0,1) to each communication edge. Although the prompt is specified, the scoring task itself is inherently subjective. Reducing complex multi-agent interactions to a ternary classification (disagreement, agreement, neutral) represents a crude simplification that may fail to capture subtle adversarial strategies. The system's reliability is fundamentally tied to the scorer LLM's ability to make consistent and accurate judgments on this ambiguous task.

W2 Computational Overhead and Scalability: While the introduction criticizes the computational overhead of baseline methods like G-Safeguard, the proposed method itself requires invoking an LLM for every edge in the graph during each evaluation round. It is reasonable to believe this could lead to significant latency and cost in large-scale MAS with extended dialogues. The paper's claims regarding efficiency lack support, as experiments are limited to small-scale MAS and provide no quantitative analysis of computational overhead.

Minor Presentation Issues: "out put" on Page 4 should be "output".

**Questions:**

The detection mechanism identifies malicious agents as statistical outliers based on their contribution scores. However, the defense operates with an inherent lag, as it requires multiple rounds of communication to compute. In a scenario where a highly contagious attack rapidly propagates and corrupts a majority of agents, could the system mistakenly identify the remaining, uncorrupted benign agents as the statistical outliers and falsely flag them as malicious? How does the method ensure robustness in such a "majority infected" scenario?

---

> ### Author Response · Authors · 2025-11-21
> **Response to Reviewer hwcL (Part 1/2)**
>
> Dear Reviewer hwcL:
>
> Thank you for recognizing our work on the originality of the dynamic defense paradigm,  the well-structured and validated method, and paper clarity. In response to your concerns, we have carefully revised the manuscript and provide detailed responses below.
>
> ---
>
> **W1**: Subjective and Coarse-Grained LLM Scoring: The core mechanism relies on an external LLM to assign a simplified ternary contribution score (-1,0,1) to each communication edge. Although the prompt is specified, the scoring task itself is inherently subjective. Reducing complex multi-agent interactions to a ternary classification (disagreement, agreement, neutral) represents a crude simplification that may fail to capture subtle adversarial strategies. The system's reliability is fundamentally tied to the scorer LLM's ability to make consistent and accurate judgments on this ambiguous task.
>
> **A1**: Thanks for your thoughtful comment. We clarify that our method does not rely solely on the ****scores ****of ****each communication edge, but also through their collaborative identification. Thus, our method still demonstrates good robustness when facing attacks where the majority of the scores are not violated. To study this circumstance where an adaptive attack infects our scoring mechanism, we conducted an attack specifically targeting the scoring process: if the original contribution value between two text segments was 1, the compromised communication node would deliberately alter the score to -1, and vice versa. The experimental results after introducing this scoring attack are as follows:
>
> *Copied from Section 4.5, Table 13: The Answer ACC against adaptive attacks on GPT-4o.*
>
> | System | Algebra | Math | Chemistry | Computer | Security | Average |
> | --- | --- | --- | --- | --- | --- | --- |
> | attack-flat | 0.79 | 0.75 | 0.65 | 0.82 | 0.81 | 0.76 |
> | attack-hierarchy | 0.81 | 0.79 | 0.68 | 0.85 | 0.8 | 0.79 |
> | **Ours-flat** | 0.92 | 0.91 | 0.73 | 0.9 | 0.83 | **0.86** |
> | **Ours-hierarchy** | 0.92 | 0.93 | 0.72 | 0.94 | 0.82 | **0.87** |
> | adaptive_attack-flat | 0.90  | 0.89 | 0.71 | 0.88 | 0.83 | 0.84  |
> | adaptive_attack-hierarchy | 0.89 | 0.91 | 0.72 | 0.92 | 0.83 | 0.85  |
>
> It can be observed that after the attack, the accuracy of our method experiences a certain decline. However, since our algorithm identifies malicious agents based on their score deviations from the majority, the scoring attack introduces variations in the scores of individual agents, but these variations are not sufficient to compromise the robustness of our approach.

---

> > ### Author Response · Authors · 2025-11-21
> > **Response to Reviewer hwcL (Part 2/2)**
> >
> > **W2**: Computational Overhead and Scalability: While the introduction criticizes the computational overhead of baseline methods like G-Safeguard, the proposed method itself requires invoking an LLM for every edge in the graph during each evaluation round. It is reasonable to believe this could lead to significant latency and cost in large-scale MAS with extended dialogues. The paper's claims regarding efficiency lack support, as experiments are limited to small-scale MAS and provide no quantitative analysis of computational overhead.
> >
> > **A2**: We investigated the growth in time consumption as the number of agents and dialogue rounds increases. Using a flat structure and the Math dataset, we extended the dialogue rounds to 5 (compared to the default setting in the paper), adding one additional round of agent suggestion and agent answer, while keeping all other settings unchanged. The time cost results from this experiment are presented below:
> >
> > Copied from Appendix B.3, Table 19: Comparison of time cost as the number of agents and dialogues gradually increases
> >
> > | Time Cost(min) |  | agent_num = 5 | agent_num = 7 | agent_num = 10 |
> > | --- | --- | --- | --- | --- |
> > | dialogue_num = 3 | flat | 1.89 | 2.86 | 4.39 |
> > | dialogue_num = 3 | Ours-flat | 2.03 | 3.11 | 4.74 |
> > | dialogue_num = 5 | flat | 3.32 | 5.04 | 7.76 |
> > | dialogue_num = 5 | Ours-flat | 3.56 | 5.35 | 8.18 |
> >
> > It can be observed that even with an increase in the number of agents and dialogue rounds, the additional time cost of our method compared to the undefended standard MAS remains modest. When the number of dialogue rounds is set to 3, our method incurs 7.41%, 8.74%, and 7.97% additional time cost for 5, 7, and 10 agents, respectively. When the number of dialogue rounds is set to 5, the additional time cost for 5, 7, and 10 agents is 7.23%, 7.34%, and 7.60%, respectively. Although the time cost growth rate of our method slightly increases with a larger number of agents, the token output required for each scoring operation remains minimal compared to the token consumption of agents when generating answers. Consequently, even as the number of agents grows, the overall time cost increase of our method remains low.
> >
> > ---
> >
> > **Q1**: The detection mechanism identifies malicious agents as statistical outliers based on their contribution scores. However, the defense operates with an inherent lag, as it requires multiple rounds of communication to compute. In a scenario where a highly contagious attack rapidly propagates and corrupts a majority of agents, could the system mistakenly identify the remaining, uncorrupted benign agents as the statistical outliers and falsely flag them as malicious? How does the method ensure robustness in such a "majority infected" scenario?
> >
> > **A3**: Thank you for raising this important point. Our method is fundamentally designed for **early detection and containment**. It operates by identifying and isolating the initial malicious agents *before* a contagion can spread to the majority, thereby preventing the scenario you described from occurring. Our experimental results across various attack types (Tables 5-8) demonstrate its high efficacy in achieving this, as the system consistently maintains robust performance by stopping attacks at their inception.
> >
> > ---
> >
> > Minor Presentation Issues: "out put" on Page 4 should be "output".
> >
> > **A4**: Thanks for your careful reading. We have corrected this typo.
> >
> > ---
> >
> > We truly appreciate your valuable and detailed feedback. If you have any further questions or concerns, please let us know.

---

### Official Review · Reviewer_KmP3 · 2025-10-30

**Soundness:** 3
**Presentation:** 2
**Contribution:** 3
**Rating:** 6
**Confidence:** 3

**Summary:**

The paper introduces a dynamic defense for LLM-based multi-agent systems: model the conversation as a time-unfolded signed DAG, back-propagate contribution scores to spot corrupted agents, and cut their edges on the fly. Experiments show it detects ≥93 % of attacks and recovers 7–10 % task accuracy, outperforming static or local baselines.

**Strengths:**

1.Node-level signed graph with back-propagation breaks the static-topology bottleneck in MAS defense.
2.Method is model-agnostic and implemented as a lightweight plug-in requiring no extra training.
3.Provides in-depth analysis of edge scoring, contribution deviation, and runtime memory overhead.

**Weaknesses:**

1. Edge-evaluation LLM may itself be biased or attacked, yet its robustness is assumed rather than tested.

2. Single fixed threshold ε for all agents/domains lacks adaptive calibration and could incur false positives/negatives.

3. Additional LLM calls per edge raise latency and cost, leaving scalability to very large agent pools unaddressed.

**Questions:**

1. Gap Shrinkage: In Table 1 and Table 5 the accuracy margin between the proposed method and the best baseline is only 1–3 pp on most cells; is this small delta sufficient to claim a clear superiority, or does it simply fall within run-to-run variance?


2. Runtime Cost: Each inference requires one extra LLM call per message for edge scoring plus a full backward pass—what is the measured latency/memory increase relative to an undefended MAS, and how does it scale as the number of agents and dialogue rounds grow?

---

> ### Author Response · Authors · 2025-11-21
> **Response to Reviewer KmP3 (Part1/4)**
>
> Dear Reviewer KmP3:
>
> Thank you for your insightful feedback and for recognizing our work on breaking the static-topology bottleneck in MAS defense, minimal computational costs and solid experimental defense performance. In response to your concerns, we have carefully revised the manuscript and provide detailed responses below.
>
> **W1**: Edge-evaluation LLM may itself be biased or attacked, yet its robustness is assumed rather than tested.
>
> **A1**: Thank you for your insightful suggestion. We acknowledge the critical role of the scoring mechanism in our experiments. To study the circumstances where an adaptive attack infects our scoring mechanism, we conducted an attack specifically targeting the scoring process: if the original contribution value between two text segments was 1, the compromised communication node would deliberately alter the score to -1, and vice versa. The experimental results after introducing this scoring attack are as follows:
>
> Copied from Section 4.5, Table 13: The Answer ACC against adaptive attacks on GPT-4o.
>
> | System | Algebra | Math | Chemistry | Computer | Security | Average |
> | --- | --- | --- | --- | --- | --- | --- |
> | attack-flat | 0.79 | 0.75 | 0.65 | 0.82 | 0.81 | 0.76 |
> | attack-hierarchy | 0.81 | 0.79 | 0.68 | 0.85 | 0.8 | 0.79 |
> | Ours-flat | 0.92 | 0.91 | 0.73 | 0.9 | 0.83 | 0.86 |
> | Ours-hierarchy | 0.92 | 0.93 | 0.72 | 0.94 | 0.82 | 0.87 |
> | adaptive_attack-flat | 0.90  | 0.89 | 0.71 | 0.88 | 0.83 | 0.84  |
> | adaptive_attack-hierarchy | 0.89 | 0.91 | 0.72 | 0.92 | 0.83 | 0.85  |
>
> It can be observed that after the attack, the accuracy of our method experiences a certain decline. However, since our algorithm identifies malicious agents based on their score deviations from the majority, the scoring attack introduces variations in the scores of individual agents, but these variations are not sufficient to compromise the robustness of our approach.
>
> ---
>
> **W2**: Single fixed threshold $\epsilon$ for all agents/domains lacks adaptive calibration and could incur false positives/negatives.
>
> **A2**: Thank you for your suggestion. In fact, $\epsilon$=1.5 is the optimal threshold we obtained after multiple experimental experiments. We conducted hyperparameter experiments on $\epsilon$ and the experimental results are as follows：
>
> Copied from Section 4.4, Table 11: The Answer ACC under different parameters on GPT-4o
>
> | System |  | Algebra | Math | Chemistry | Computer | Security | Average |
> | --- | --- | --- | --- | --- | --- | --- | --- |
> | $\epsilon$=1.3 | flat | 0.91 | 0.89 | 0.69 | 0.88 | 0.81 | 0.84 |
> |  | hierarchy | 0.90 | 0.90 | 0.69 | 0.86 | 0.80 | 0.83 |
> | $\epsilon$=1.4 | flat | 0.92 | 0.89 | 0.71 | 0.88 | 0.82 | 0.84 |
> |  | hierarchy | 0.90 | 0.91 | 0.70 | 0.87 | 0.81 | 0.84 |
> | $\epsilon$=1.5 | flat | 0.92 | 0.91 | 0.73 | 0.9 | 0.83 | 0.86 |
> |  | hierarchy | 0.92 | 0.93 | 0.72 | 0.94 | 0.82 | 0.87 |
> | $\epsilon$=1.6 | flat | 0.88 | 0.88 | 0.70 | 0.87 | 0.83 | 0.83 |
> |  | hierarchy | 0.9 | 0.85 | 0.71 | 0.91 | 0.81 | 0.84 |
> | $\epsilon$=1.7 | flat | 0.85 | 0.81 | 0.67 | 0.84 | 0.82 | 0.80 |
> |  | hierarchy | 0.87 | 0.85 | 0.69 | 0.88 | 0.81 | 0.82 |
>
> Please note that we have applied normalization when calculating the score for each agent. Even if the structure of the agent graph changes, the threshold does not fluctuate significantly. It can be observed that our method performs best at $\epsilon$=1.5 for both flat and hierarchical system architectures. Furthermore, by constraining the scoring options of the evaluation model to {-1, 0, 1}, the model scores remain consistent across different datasets. In summary, the scoring of our method is robust across different architectures and datasets, resulting in high threshold robustness.

---

> ### Author Response · Authors · 2025-11-21
> **Response to Reviewer KmP3 (Part 2/4)**
>
> **W3**: Additional LLM calls per edge raise latency and cost, leaving scalability to very large agent pools unaddressed.
>
> **A3**: Thank you for your suggestion. We have conducted experiments to measure the time cost of our method by comparing the average time required to complete tasks between our approach and a standard MAS. The results are summarized in the table below:
>
> *Copied from Appendix B.3, Table 18: Comparison of Time Cost between Our Method and Standard MAS Method*
>
> | Time Cost(min) | Algebra | Math | Chemistry | Computer | Security | Average |
> | --- | --- | --- | --- | --- | --- | --- |
> | flat | 2.03 | 1.89 | 1.68 | 1.76 | 1.52 | 1.776 |
> | hierarchy | 1.13 | 0.98 | 0.89 | 0.94 | 0.87 | 0.962 |
> | Ours-flat | 2.21 | 2.03 | 1.85 | 1.91 | 1.68 | 1.936 |
> | Ours-hierarchy | 1.21 | 1.14 | 0.98 | 1.03 | 0.96 | 1.064 |
>
> It can be observed that the average time consumption of the standard MAS under flat and hierarchical structures is 1.776 minutes and 0.962 minutes, respectively. In comparison, our method introduces **only an additional 9.0% and 10.6% time overhead** for the flat and hierarchical structures, respectively. This marginal increase is attributed to the fact that the external evaluator only scores pairs of sentences, and the token consumption for this process is negligible compared to that required for answering the original questions. Even with an increase in the number of agents, the time required for the evaluator to perform scoring does not grow significantly. Therefore, our method incurs minimal additional cost while ensuring the security of the MAS system.
>
> In addition, we investigated the growth in time consumption as the number of agents and dialogue rounds increases. Using a flat structure and the Math dataset, we extended the dialogue rounds to 5 (compared to the default setting in the paper), adding one additional round of agent suggestion and agent answer, while keeping all other settings unchanged. The time cost results from this experiment are presented below:
>
> *Copied from Appendix B.3, Table 19: Comparison of time cost as the number of agents and dialogues gradually increases*
>
> | Time Cost(min) |  | agent_num = 5 | agent_num = 7 | agent_num = 10 |
> | --- | --- | --- | --- | --- |
> | dialogue_num = 3 | flat | 1.89 | 2.86 | 4.39 |
> | dialogue_num = 3 | Ours-flat | 2.03 | 3.11 | 4.74 |
> | dialogue_num = 5 | flat | 3.32 | 5.04 | 7.76 |
> | dialogue_num = 5 | Ours-flat | 3.56 | 5.35 | 8.18 |
>
> It can be observed that even with an increase in the number of agents and dialogue rounds, the additional time cost of our method compared to the undefended standard MAS remains modest. When the number of dialogue rounds is set to 3, our method incurs 7.41%, 8.74%, and 7.97% additional time cost for 5, 7, and 10 agents, respectively. When the number of dialogue rounds is set to 5, the additional time cost for 5, 7, and 10 agents is 7.23%, 7.34%, and 7.60%, respectively. Therefore, the overhead is still admissible as the number of agents and dialogue rounds increases.

---

> > ### Author Response · Authors · 2025-11-21
> > **Response to Reviewer KmP3 (Part 3/4)**
> >
> > **A4**: Thank you for your suggestion. We conducted 3 experiments on Table 1 of the main experiment and calculated 1 standard deviation in each experiment. We updated them into Table 1, and the results are as follows：
> >
> > Copied from Section 4.2, Table 1: The Answer ACC of different methods on the GPT-4o model using the MMLU dataset. We conduct 3 experiments and present 1 standard deviation in each experiment.
> >
> > | System | Algebra | Math | Chemistry | Computer | Security | \textbf{Average} |
> > | --- | --- | --- | --- | --- | --- | --- |
> > | single | 0.897 ± 0.006 | 0.887 ± 0.006 | 0.707 ± 0.015 | 0.867 ± 0.012 | 0.807 ± 0.012 | 0.833 ± 0.006 |
> > | flat | 0.95 ± 0.01 | 0.947 ± 0.012 | 0.753 ± 0.006 | 0.92 ± 0.01 | 0.85 ± 0.01 | 0.884 ± 0.006 |
> > | hierarchy | 0.957 ± 0.015 | 0.953 ± 0.015 | 0.753 ± 0.025 | 0.95 ± 0.01 | 0.823 ± 0.006 | 0.887 ± 0.01 |
> > | attack-flat | 0.787 ± 0.006 | 0.747 ± 0.006 | 0.647 ± 0.006 | 0.823 ± 0.015 | 0.81 ± 0.01 | 0.763 ± 0.006 |
> > | attack-hierarchy | 0.817 ± 0.006 | 0.817 ± 0.025 | 0.667 ± 0.015 | 0.847 ± 0.006 | 0.78 ± 0.017 | 0.786 ± 0.006 |
> > | Ours-flat | 0.923 ± 0.006 | 0.93 ± 0.017 | 0.733 ± 0.015 | 0.873 ± 0.023 | 0.81 ± 0.02 | 0.854 ± 0.006 |
> > | Ours-hierarchy | 0.933 ± 0.012 | 0.957 ± 0.025 | 0.737 ± 0.015 | 0.91 ± 0.03 | 0.837 ± 0.015 | 0.875 ± 0.006 |
> > | G-Safeguard-flat | 0.883 ± 0.006 | 0.887 ± 0.006 | 0.71 ± 0.02 | 0.877 ± 0.015 | 0.83 ± 0.02 | 0.837 ± 0.01 |
> > | G-Safeguard-hierarchy | 0.92 ± 0.01 | 0.917 ± 0.021 | 0.713 ± 0.015 | 0.903 ± 0.023 | 0.823 ± 0.012 | 0.855 ± 0.006 |
> > | AGENTXPOSED-flat | 0.9 ± 0.01 | 0.79 ± 0.01 | 0.67 ± 0.01 | 0.89 ± 0.017 | 0.873 ± 0.015 | 0.825 ± 0.012 |
> > | AGENTXPOSED-hierarchy | 0.853 ± 0.006 | 0.833 ± 0.012 | 0.687 ± 0.006 | 0.9 ± 0.01 | 0.827 ± 0.015 | 0.82 ± 0 |
> > | Challenger-flat | 0.887 ± 0.006 | 0.873 ± 0.012 | 0.683 ± 0.006 | 0.84 ± 0.017 | 0.753 ± 0.025 | 0.807 ± 0.006 |
> > | Challenger-hierarchy | 0.863 ± 0.006 | 0.873 ± 0.012 | 0.677 ± 0.006 | 0.863 ± 0.015 | 0.783 ± 0.012 | 0.812 ± 0.012 |
> > | Inspector-flat | 0.84 ± 0.01 | 0.89 ± 0.01 | 0.657 ± 0.006 | 0.807 ± 0.015 | 0.757 ± 0.021 | 0.79 ± 0.012 |
> > | Inspector-hierarchy | 0.87 ± 0.01 | 0.897 ± 0.015 | 0.683 ± 0.006 | 0.88 ± 0.017 | 0.79 ± 0.017 | 0.824 ± 0.01 |
> >
> > It can be observed that the superiority of our method over other baselines is not due to chance. The standard deviation of our method is only 0.006 in both flat and hierarchical structures, indicating highly stable performance. Meanwhile, the average accuracy of our method reaches 0.854 and 0.875 in the two structures, respectively, demonstrating a significant improvement over the attacked scenarios (0.763 and 0.786) and a substantial advantage over the best baseline (0.837 and 0.855). These results fully validate the stability and effectiveness of our proposed method.

---

> > > ### Author Response · Authors · 2025-11-21
> > > **Response to Reviewer KmP3 (Part 4/4)**
> > >
> > > **Q2:** Runtime Cost: Each inference requires one extra LLM call per message for edge scoring plus a full backward pass—what is the measured latency/memory increase relative to an undefended MAS, and how does it scale as the number of agents and dialogue rounds grow?
> > >
> > > **A5:** Thank you for your suggestion. We have conducted experiments to measure the time cost of our method by comparing the average time required to complete tasks between our approach and a standard MAS. The results are summarized in the table below:
> > >
> > > *Copied from Appendix B.3, Table 18: Comparison of Time Cost between Our Method and Standard MAS Method*
> > >
> > > | Time Cost(min) | Algebra | Math | Chemistry | Computer | Security | Average |
> > > | --- | --- | --- | --- | --- | --- | --- |
> > > | flat | 2.03 | 1.89 | 1.68 | 1.76 | 1.52 | 1.776 |
> > > | hierarchy | 1.13 | 0.98 | 0.89 | 0.94 | 0.87 | 0.962 |
> > > | Ours-flat | 2.21 | 2.03 | 1.85 | 1.91 | 1.68 | 1.936 |
> > > | Ours-hierarchy | 1.21 | 1.14 | 0.98 | 1.03 | 0.96 | 1.064 |
> > >
> > > It can be observed that the average time consumption of the standard MAS under flat and hierarchical structures is 1.776 minutes and 0.962 minutes, respectively. In comparison, our method introduces only an additional 9.0% and 10.6% time overhead for the flat and hierarchical structures, respectively. This marginal increase is attributed to the fact that the external evaluator only scores pairs of sentences, and the token consumption for this process is negligible compared to that required for answering the original questions. Even with an increase in the number of agents, the time required for the evaluator to perform scoring does not grow significantly. Therefore, our method incurs minimal additional cost while ensuring the security of the MAS system.
> > >
> > > In addition, we investigated the growth in time consumption as the number of agents and dialogue rounds increases. Using a flat structure and the Math dataset, we extended the dialogue rounds to 5 (compared to the default setting in the paper), adding one additional round of agent suggestion and agent answer, while keeping all other settings unchanged. The time cost results from this experiment are presented below:
> > >
> > > *Copied from Appendix B.3, Table 19: Comparison of time cost as the number of agents and dialogues gradually increases*
> > >
> > > | Time Cost(min) |  | agent_num = 5 | agent_num = 7 | agent_num = 10 |
> > > | --- | --- | --- | --- | --- |
> > > | dialogue_num = 3 | flat | 1.89 | 2.86 | 4.39 |
> > > | dialogue_num = 3 | Ours-flat | 2.03 | 3.11 | 4.74 |
> > > | dialogue_num = 5 | flat | 3.32 | 5.04 | 7.76 |
> > > | dialogue_num = 5 | Ours-flat | 3.56 | 5.35 | 8.18 |
> > >
> > > It can be observed that even with an increase in the number of agents and dialogue rounds, the additional time cost of our method compared to the undefended standard MAS remains modest. When the number of dialogue rounds is set to 3, our method incurs 7.41%, 8.74%, and 7.97% additional time cost for 5, 7, and 10 agents, respectively. When the number of dialogue rounds is set to 5, the additional time cost for 5, 7, and 10 agents is 7.23%, 7.34%, and 7.60%, respectively.
> > >
> > > ---
> > >
> > > We truly appreciate your valuable and detailed feedback. If you have any further questions or concerns, please let us know.

---

### Official Review · Reviewer_3kEQ · 2025-11-03

**Soundness:** 2
**Presentation:** 3
**Contribution:** 2
**Rating:** 4
**Confidence:** 3

**Summary:**

The paper addresses security in LLM-based multi-agent systems by treating agent interactions as a time-unrolled DAG and scoring each message with an external LLM to see whether it helps or harms the final task. These local edge scores are then propagated backward through the graph so every agent gets a global contribution score, letting the system spot agents whose behavior consistently deviates. Detected agents have their outgoing edges cut, giving a dynamic defense that adapts as attacks evolve rather than relying on a fixed topology. Experiments across tasks, LLMs, and attack types show this beats prior MAS defenses, especially on subtle “modification” attacks.

**Strengths:**

1. propose to model evolving agent graphs instead of fixed topology.
2. Graph backpropagation mechanism for attribution. Turning per-edge LLM judgments into a global contribution score via backward propagation over the DAG is a neat, principled way to answer “which agent actually pushed the system off course?”.
3. Broad, heterogenous evaluation. They test across multiple MAS structures, LLM backbones, tasks, and attack styles, and show consistent gains over G-Safeguard / AgentXposed / Inspector, which strengthens the generality claim.

**Weaknesses:**

1. LLM-as-judge dependency. The whole pipeline leans on an external LLM to label edge contributions (−1,0,1); if that judge is noisy or biased, the attribution and thus the defense can collapse. Also, if the attacker is steathy or lauched gailbreak against this LLM, the defense could aslo break.
2. Missing computation overhead of maintaining and backproping dynamically changing graphs.
3. Missing analysis over the hyperparam threshold

**Questions:**

Please see Weaknesses

---

> ### Author Response · Authors · 2025-11-21
> **Response to Reviewer 3kEQ (Part 1/3)**
>
> Dear Reviewer 3kEQ,
>
> Thank you for acknowledging our work on the novel graph back propagation method, applicability to model evolving agent graphs, and experimental MAS defense performance.  In response to your concerns, we have carefully revised the manuscript and provide detailed responses below.
>
> ---
>
> **Q1**: LLM-as-judge dependency. The whole pipeline leans on an external LLM to label edge contributions (−1,0,1); if that judge is noisy or biased, the attribution and thus the defense can collapse. Also, if the attacker is stealthy or launched jailbreak against this LLM, the defense could also break.
>
> **A1**: Thank you for your insightful suggestion. We acknowledge the critical role of the scoring mechanism in our experiments. To study the circumstances where an adaptive attack infects our scoring mechanism, **we have designed an adaptive attack** specifically targeting the scoring process: if the original contribution value between two text segments was 1, the compromised communication node would deliberately alter the score to -1, and vice versa. The experimental results after introducing this scoring attack are as follows:
>
> *Copied from Section 4.5, Table 13: The Answer ACC against adaptive attacks on GPT-4o*
>
> | System | Algebra | Math | Chemistry | Computer | Security | Average |
> | --- | --- | --- | --- | --- | --- | --- |
> | attack-flat | 0.79 | 0.75 | 0.65 | 0.82 | 0.81 | 0.76 |
> | attack-hierarchy | 0.81 | 0.79 | 0.68 | 0.85 | 0.8 | 0.79 |
> | **Ours-flat** | 0.92 | 0.91 | 0.73 | 0.9 | 0.83 | **0.86** |
> | **Ours-hierarchy** | 0.92 | 0.93 | 0.72 | 0.94 | 0.82 | **0.87** |
> | adaptive_attack-flat | 0.90  | 0.89 | 0.71 | 0.88 | 0.83 | 0.84  |
> | adaptive_attack-hierarchy | 0.89 | 0.91 | 0.72 | 0.92 | 0.83 | 0.85  |
>
> It can be observed that after the adaptive attack, the accuracy of our method experiences a certain decline. However, since our algorithm identifies malicious agents based on their score deviations from the majority, the scoring attack introduces variations in the scores of individual agents, but these variations are not sufficient to compromise the robustness of our approach.
>
> Thanks again for the valuable suggestion!
>
> ---

---

> > ### Author Response · Authors · 2025-11-21
> > **Response to Reviewer 3kEQ (Part 2/3)**
> >
> > **Q2**: Missing computation overhead of maintaining and backpropagating dynamically changing graphs.
> > **A2**: Thank you for your careful reading. We have conducted experiments to measure the time cost of our method by comparing the average time required to complete tasks between our approach and a standard MAS. The results are summarized in the table below:
> >
> > *Table Appendix B.3, 18: Comparison of Time Cost between Our Method and the Standard MAS Method*
> >
> > | Time Cost(min) | Algebra | Math | Chemistry | Computer | Security | Average |
> > | --- | --- | --- | --- | --- | --- | --- |
> > | flat | 2.03 | 1.89 | 1.68 | 1.76 | 1.52 | 1.776 |
> > | hierarchy | 1.13 | 0.98 | 0.89 | 0.94 | 0.87 | 0.962 |
> > | **Ours-flat** | 2.21 | 2.03 | 1.85 | 1.91 | 1.68 | 1.936 |
> > | **Ours-hierarchy** | 1.21 | 1.14 | 0.98 | 1.03 | 0.96 | 1.064 |
> >
> > It can be observed that the average time consumption of the standard MAS under flat and hierarchical structures is 1.776 minutes and 0.962 minutes, respectively. In comparison, our method introduces only an additional 9.0% and 10.6% time overhead for the flat and hierarchical structures, respectively. This marginal increase is attributed to the fact that the external evaluator only scores pairs of sentences, and the token consumption for this process is negligible compared to that required for answering the original questions. Even with an increase in the number of agents, the time required for the evaluator to perform scoring does not grow significantly. Therefore, our method incurs minimal additional cost while ensuring the security of the MAS system.
> >
> > In addition, we investigated the growth in time consumption as the number of agents and dialogue rounds increases. Using a flat structure and the Math dataset, we extended the dialogue rounds to 5 (compared to the default setting in the paper), adding one additional round of agent suggestion and agent answer, while keeping all other settings unchanged. The time cost results from this experiment are presented below:
> >
> > *Copied from Appendix B.3, Table 19: Comparison of time cost as the number of agents and dialogues gradually increases.*
> >
> > | Time Cost(min) |  | agent_num = 5 | agent_num = 7 | agent_num = 10 |
> > | --- | --- | --- | --- | --- |
> > | dialogue_num = 3 | flat | 1.89 | 2.86 | 4.39 |
> > |  | Ours-flat | 2.03 | 3.11 | 4.74 |
> > | dialogue_num = 5 | flat | 3.32 | 5.04 | 7.76 |
> > |  | Ours-flat | 3.56 | 5.35 | 8.18 |
> >
> > It can be observed that even with an increase in the number of agents and dialogue rounds, the additional time cost of our method compared to the undefended standard MAS remains modest. When the number of dialogue rounds is set to 3, our method incurs 7.41%, 8.74%, and 7.97% additional time cost for 5, 7, and 10 agents, respectively. When the number of dialogue rounds is set to 5, the additional time cost for 5, 7, and 10 agents is 7.23%, 7.34%, and 7.60%, respectively.
> >
> > ---

---

> > > ### Author Response · Authors · 2025-11-21
> > > **Response to Reviewer 3kEQ (Part 3/3)**
> > >
> > > **Q3**: Missing analysis over the hyperparam threshold
> > > **A3**: We thank the reviewer for this suggestion. The threshold $\epsilon$ in Eq. 2 is used to identify malicious agents based on score deviation. In our experiments, we set $\epsilon$ = 1.5 based on a grid search over a validation set, which consistently yielded the best balance between precision and recall. We have compiled the adjustment results of $\epsilon$ in our study into ablation experiments, and the experimental results are as follows:
> > >
> > > *Copied from Section 4.4, Table 11: The Answer ACC under different parameters on GPT-4o*
> > >
> > > | System |  | Algebra | Math | Chemistry | Computer | Security | Average |
> > > | --- | --- | --- | --- | --- | --- | --- | --- |
> > > | $\epsilon$=1.3 | flat | 0.91 | 0.89 | 0.69 | 0.88 | 0.81 | 0.84 |
> > > |  | hierarchy | 0.90 | 0.90 | 0.69 | 0.86 | 0.80 | 0.83 |
> > > | $\epsilon$=1.4 | flat | 0.92 | 0.89 | 0.71 | 0.88 | 0.82 | 0.84 |
> > > |  | hierarchy | 0.90 | 0.91 | 0.70 | 0.87 | 0.81 | 0.84 |
> > > | $\epsilon$=1.5 | flat | 0.92 | 0.91 | 0.73 | 0.9 | 0.83 | 0.86 |
> > > |  | hierarchy | 0.92 | 0.93 | 0.72 | 0.94 | 0.82 | 0.87 |
> > > | $\epsilon$=1.6 | flat | 0.88 | 0.88 | 0.70 | 0.87 | 0.83 | 0.83 |
> > > |  | hierarchy | 0.9 | 0.85 | 0.71 | 0.91 | 0.81 | 0.84 |
> > > | $\epsilon$=1.7 | flat | 0.85 | 0.81 | 0.67 | 0.84 | 0.82 | 0.80 |
> > > |  | hierarchy | 0.87 | 0.85 | 0.69 | 0.88 | 0.81 | 0.82 |
> > >
> > > *Copied from Section 4.4, Table 12: The Monitor ACC under different parameters on GPT-4o*
> > >
> > > | System |  | Algebra | Math | Chemistry | Computer | Security | Average |
> > > | --- | --- | --- | --- | --- | --- | --- | --- |
> > > | $\epsilon$=1.3 | flat | 0.98 | 0.97 | 0.94 | 0.95 | 0.94 | 0.96 |
> > > |  | hierarchy | 0.97 | 0.96 | 0.97 | 0.94 | 0.97 | 0.96 |
> > > | $\epsilon$=1.4 | flat | 0.96 | 0.93 | 0.93 | 0.91 | 0.91 | 0.93 |
> > > |  | hierarchy | 0.94 | 0.95 | 0.92 | 0.89 | 0.92 | 0.92 |
> > > | $\epsilon$=1.5 | flat | 0.91 | 0.89 | 0.85 | 0.85 | 0.86 | 0.87 |
> > > |  | hierarchy | 0.92 | 0.91 | 0.87 | 0.86 | 0.87 | 0.89 |
> > > | $\epsilon$=1.6 | flat | 0.82 | 0.78 | 0.76 | 0.73 | 0.71 | 0.76 |
> > > |  | hierarchy | 0.75 | 0.77 | 0.74 | 0.69 | 0.65 | 0.72 |
> > > | $\epsilon$=1.7 | flat | 0.63 | 0.54 | 0.52 | 0.41 | 0.43 | 0.51 |
> > > |  | hierarchy | 0.49 | 0.47 | 0.46 | 0.38 | 0.34 | 0.43 |
> > >
> > > As $\epsilon$ increases from 1.3 to 1.5, the average accuracy of the MAS across various tasks improves from 0.84 to 0.87. However, when $\epsilon$ further increases from 1.5 to 1.7, the average accuracy drops to 0.82, indicating that $\epsilon$ = 1.5 is the optimal hyperparameter. This phenomenon can be explained by the detection accuracy of malicious agents. When $\epsilon$ is relatively small, the lower threshold leads to more agents being identified as malicious, thereby increasing the detection rate from 0.88 to 0.96. However, this also results in the removal of certain valid communication links during the subsequent edge-pruning process, which slightly degrades the overall accuracy. Conversely, when $\epsilon$ is too large, the higher threshold prevents the detection of some malicious agents, reducing the detection rate from 0.88 to 0.47. As a result, the harmful attacks persist, leading to a decline in the MAS's task performance accuracy.

---

### Author Response · Authors · 2025-12-01
**Final Remarks for Area Chair**

Dear Area Chair,

We sincerely appreciate your intervention in overseeing the review process of our submission under these special circumstances. We understand the significant additional workload brought about by this reassignment and are grateful for the time and effort you have invested in evaluating our work, the original reviews, and the rebuttal updates.

---

**Summary of Strengths**

The reviewers highlighted the following main strengths of our work:

- **Novel Graph Modeling and Propagation Mechanism:** Modeling the multi-agent system as a time-unrolled directed acyclic graph and evaluating node contributions through a backpropagation mechanism breaks the static topology bottleneck (Reviewer 3kEQ, KmP3, hwcL).
- **Extensive Experimental Validation:** Comprehensive evaluation across various MAS architectures, tasks, attack types, and LLM backbones demonstrates the method's generality and robustness (Reviewer 3kEQ, KmP3).
- **Model-Agnosticism and Lightweight Implementation:** The method requires no additional training and can be integrated as a plug-in into existing systems, offering good practicality (Reviewer KmP3).
- **Clear Paper Structure and Logical Flow:** The paper is clearly written, with well-organized figures and numerical results, making it easy to follow (Reviewer hwcL).

---

**Responses to Key Concerns**

1. **Dependency and Robustness of the LLM Scoring Mechanism Concern:** Reviewers expressed concerns that the external LLM scorer could be noisy, biased, or attacked, thereby affecting the overall defense effectiveness (Reviewer 3kEQ, KmP3, hwcL).

**Solution:** We have designed an adaptive attack experiment for the scoring process： if the original contribution value between two text segments was 1, the compromised communication node would deliberately alter the score to -1, and vice versa. The results indicate that although there is a slight decrease in performance, the variation in individual scores is not sufficient to compromise the robustness of our method Therefore, our method still maintains a good defense.

**Location:** (Newly revised) Section 4.5, Table 13.


2. **Lack of Computational Overhead and Scalability Analysis Concern:** Reviewers noted the lack of quantitative analysis regarding the computational overhead of maintaining and backpropagating through dynamically changing graphs (Reviewer 3kEQ, KmP3, hwcL).

**Solution:** We conducted additional time cost experiments, and the results showed that under default settings, our method only increased the additional time overhead by about 9-10% in both flat and layered structures; In addition, even if the number of agents increases to 7 and 10, and the number of conversation rounds increases to 5, the additional time cost of our method is still controlled within 10%.

**Location:** (Newly revised) Appendix B.3, Tables 18 and 19.


3. **Insufficient Analysis of the Hyperparameter Threshold Concern:** Reviewers pointed out that the threshold ε lacked adaptive calibration and could lead to false positives/negatives (Reviewer 3kEQ, KmP3).

**Solution:** We provided ablation studies for ε ranging from 1.3 to 1.7, the experiment result showing that: as ε increases from 1.3 to 1.5, the average accuracy of the MAS across various tasks improves from 0.84 to 0.87. However, when ε further increases from 1.5 to 1.7, the average accuracy drops to 0.82, indicating that ε = 1.5 is the optimal hyperparameter.

**Location:** (Newly revised) Section 4.4, Tables 11 and 12.


4. **Statistical Significance of Performance Advantage Concern:** Reviewers questioned whether the performance improvement compared to baseline methods was significant (Reviewer KmP3).

**Solution:** We supplemented the analysis with standard deviations from three repeated experiments. The experiment showing that: the standard deviation of our method is only 0.006 in both flat and hierarchical structures. Meanwhile, the average accuracy of our method(0.854 and 0.875) demonstrating a substantial advantage over the best baseline (0.837 and 0.855).

**Location:** (Newly revised) Section 4.2, Table 1.


5. **Robustness in Majority-Infected Scenarios Concern:** Reviewers were concerned that in scenarios where a majority of agents are infected, benign nodes might be misidentified as outliers (Reviewer hwcL).

**Solution:** We emphasize that this method is designed for early detection and containment. The experimental results (Table 5-8) demonstrate its effectiveness in preventing the spread of attacks in the early stages and preventing the occurrence of most infection scenarios. Therefore, our method does not have the risk of most agents being infected.

**Location:** (Original content) Sections 4.2–4.3, Tables 5–8.

---

### Meta-Review · Area_Chair_XETC · 2026-01-01

**Summary:**

The reviewers agree that the paper proposes an interesting and well-motivated approach for defending LLM-based multi-agent systems by modeling interactions as a dynamic graph and attributing agent influence via backward propagation. The main concerns are about the proposed method's reliability and practical impact. In particular, reviewers questioned the dependence on an external LLM as a judge, the limited adaptivity of the thresholding mechanism, and whether the reported gains over baselines are large enough to justify the added complexity and cost. While the experimental section is extensive, several reviewers felt that the improvements are often modest and that key assumptions remain insufficiently validated, especially for large-scale or adversarially adaptive settings.

**Reviewer Concerns:**

### Concerns addressed in the rebuttal: ###

- The authors added experiments analyzing the robustness of the method when the LLM-based edge scorer is attacked, partially addressing concerns raised by Reviewer 3kEQ and Reviewer KmP3 about reliance on an external judge.

- Additional ablations and runtime measurements were provided to respond to questions about computational overhead and the choice of the threshold parameter, which addressed some of the clarity gaps noted by Reviewer 3kEQ and Reviewer KmP3.

- Variance reporting helped clarify that some performance gains are statistically stable, responding to Reviewer KmP3’s question about whether improvements fall within noise.

### Concerns that remain outstanding: ###

- Despite added experiments, it remained unconvinced that robustness to evaluator compromise is fully established, especially against stronger or more realistic adaptive attacks.

- The use of a single global threshold is still heuristic, and the rebuttal does not fully resolve concerns about domain- or scale-dependent behavior.

- The practical impact remains unclear, where several reviewers noted that accuracy gains over strong baselines are often small, raising doubts about whether the added latency, cost, and system complexity are justified in real deployments.

**Reviewer Scores:**

- Reviewer 3kEQ: Original score 4. After rebuttal, this reviewer would likely remain at 4, as their core concerns about robustness and assumptions were only partially addressed.

- Reviewer KmP3: Original score 6. With the additional experiments, this reviewer might still stay at 6, but not increase.

- Reviewer hwcL: Original score 8. Based on their emphasis on failure cases and majority-infected scenarios, their stance would likely remain unchanged.

---

### Decision · Program_Chairs · 2026-01-26

Reject